# OpenReview forum: "Graph Neural Dynamics via Learned Energy and Tangential Flows"
_ICML.cc/2026/Conference — ICML 2026 regular_

### Official Review · Reviewer_BFpR · 2026-03-10

**Soundness:** 3
**Presentation:** 3
**Significance:** 3
**Originality:** 3
**Overall Recommendation:** 5
**Confidence:** 3

**Summary:**

This paper proposes a framework, TANGO, that learns an energy function to solve a downstream task on graphs, and defines GNN layer updates containing two terms: one energy minimization step via the energy gradient and one tangent step via the tangent direction.
The framework is theoretically characterized to show its advantages, namely the capability to move out of flat areas and learn quadratic updates, and is extensively evaluated on various real-world benchmarks.

**Compliance With Llm Reviewing Policy:**

Affirmed.

**Final Justification:**

The authors' rebuttal clarified my concerns. I have increased my score accordingly.

**Key Questions For Authors:**

1. Is there a motivation why $\alpha_G$ and $\beta_G$ are learned instead of being, e.g., hyperparameters?
2. How does Proposition 4.1 impact the analysis of feature evolution in TANGO?
3. Does the implementation of EnergyGNN and TangentGNN affect the properties of TANGO (e.g., if they are deep GNNs suffering from feature collapse)? Are there any guidelines or restrictions on their implementation?

**Limitations:**

Yes

**Strengths And Weaknesses:**

**Strengths:**
- The paper presents a thorough evaluation on real datasets and ablation studies
- The proposed method is supported by theoretical results
- The presentation is clear and easy to follow

**Weaknesses:**
- The intuition for adding a tangential descent direction could be better motivated. While it is true that it may avoid being stuck in flat surfaces, it might as well pull away from optima and there is no guarantee it will identify a better update direction than pure gradient updates. The theoretical results describe potential behaviors (it *can* evolve in energy-flat regions, it *can* learn Newton directions) but there is no guarantee nor evaluation to confirm whether this happens or not, and the performance improvements in experiments might be due to other factors. A controlled synthetic experiment where the learned energy function and the descent/tangential directions are analyzed in depth would help clarify this behavior.
- The implications of Proposition 4.1 could be better discussed. The authors state that, as a consequence of the energy being non-increasing, the solutions are not forced to collapse in feature space. I believe this connection is not immediate and would benefit from a clearer explanation.
- The practical implementation of the framework ($\alpha_G$, $\beta_G$, EnergyGNN, TangentGNN) could be better justified.

---

> ### Author Rebuttal · Authors · 2026-03-31
>
> We sincerely thank you for highlighting the strong empirical evaluation, theoretical support, and clear presentation of the paper. We are grateful for your constructive questions: we found them to help us highlight the motivation TANGO. We hope that you find our responses satisfactory, and that you will consider revising your score.
>
> ---
>
> **1. Re tangential direction intuition:** Thank you for this important comment.  In continuous time, it is orthogonal to $\nabla_H V_G$, so $\frac{d}{dt}V_G(H(t))=-\alpha_G(H(t))|\nabla_H V_G(H(t))|^2$: by design, mathematically, it does not drive the dynamics uphill. Rather, it enables motion along level sets while the descent term controls dissipation. This is especially useful in flat or ill-conditioned regimes, where pure gradient flow can be weak, as shown in our paper. Our Section 4 results are capacity-based results, which is a common approach in machine and graph learning [1–3], characterizing what TANGO can realize relative to pure gradient-flow updates. The experiments test the performance and contribution of TANGO in real-life scenarios.
>
> To further accommodate your comment on the tangential update, we add a controlled synthetic experiment on the barbell graph as in Figure 2 that isolates the tangential term from other factors. We directly compare matched one-step updates from the same visited states: gradient-flow vs. TANGO. We measure changes in task loss, cross-clique transmission, and the alignment $\cos(d_{\mathrm{tan}},g_L^{\mathrm{tan}})$. The results in the table below show that the learned tangential direction is positively aligned with the loss-reducing tangent direction, and that TANGO yields substantially larger one-step gains than pure gradient descent in flat-energy regimes, where gradient descent alone makes almost no progress, while still remaining beneficial in low-loss states rather than pulling away from optima. This directly addresses the concern by showing, in a controlled setting, that the tangential term contributes useful directions beyond pure gradient descent and that the gains are not due to unrelated architectural factors. Thank you for the insightful suggestion for an experiment, which we also added to our revised paper, and in our opinion improves the understanding of our approach.
>
> | Regime | $\cos(d_{\mathrm{tan}},g_L^{\mathrm{tan}})$ | $\Delta$Loss, gradient only | $\Delta$Loss, TANGO | $\Delta$Transmission, gradient only | $\Delta$Transmission, TANGO |
> |---|---:|---:|---:|---:|---:|
> | flat-energy states | 0.2857 | -0.0005 | -0.0904 | +0.0006 | +0.1105 |
> | low-loss states | 0.2339 | -0.00003 | -0.0103 | +0.00005 | +0.0131 |
>
> [1] Universal Approximation with Deep Narrow Networks, COLT 2020.
>
> [2]How Powerful are Graph Neural Networks?, ICLR 2019.
>
> [3] Generalization, Expressivity, and Universality of Graph Neural Networks on Attributed Graphs, ICLR 2025.
>
> ---
>
> **2. Re Proposition 4.1 and feature evolution:** Thank you for the comment. The proposition shows that the learned energy is non-increasing and bounded below, so the trajectory remains in ${H: V_G(H)\leq V_G(H^{(0)})}$. Since the descent coefficient is task-driven, that is, learned, it may also reduce or stop descent. Thus, Proposition 4.1 is a stability statement about the energy, and our intention was to clarify that it does not cause feature collapse. Moreover, in TANGO, the tangential component explicitly allows motion along these level sets, so the dynamics remain flexible under Lyapunov control. This is consistent with Proposition 4.2, which shows that when $\nabla_H V_G(H)=0$ and $T_{V_G}(H)\neq 0$, the features can still evolve through the tangential term. We revised the paper to clarify this point. Thank you.
>
> ---
>
> **3. Re learning coefficients:** The motivation is to make TANGO adaptive. Fixed coefficients would impose the same balance between descent and tangential motion across all graphs, layers, and stages of feature evolution. Instead, TANGO predicts $\alpha_G$ and $\beta_G$ from the hidden features, allowing the model to choose the balance based on the current state. We made this rationale more explicit in the revised paper. Thank you.
>
> ---
>
> **4.  Re EnergyGNN, TangentGNN, $\alpha_G$ and $\beta_G$:** TANGO is architecture-agnostic at the framework level: its core properties arise from the learned scalar energy, gradient term, and orthogonal tangential projection, while the backbone contributes its own inductive biases, as in any GNN framework. This is why cross-backbone evaluation is important, and our experiments show consistent performance gains over GCN, GIN, GatedGCN, and GPS. As with any backbone-based method, TANGO is best paired with a sound base network (e.g., a non collapsing backbone), and our results show that it couples well with diverse backbones and consistently improves their performance. We revised the paper to clarify this point. Thank you.

---

> > ### Author Rebuttal · Reviewer_BFpR · 2026-04-01
> >
> > I thank the authors for their clarifications and for the synthetic experiment, which addressed my concerns. I have increased my score.

---

> > > ### Author Response · Authors · 2026-04-04
> > >
> > > Dear Reviewer BFpR,
> > >
> > > We thank you for your thoughtful and actionable review, and for your rebuttal acknowledgement. We found your feedback to be beneficial to the quality of our work.
> > >
> > > We appreciate your acknowledgment of satisfaction with our rebuttal, stating that it fully resolves your concerns. We would like to thank you for increasing your score and recommending the acceptance of our work.
> > >
> > >
> > > With best regards, \
> > > Authors.

---

### Official Review · Reviewer_YJEt · 2026-03-13

**Soundness:** 3
**Presentation:** 3
**Significance:** 3
**Originality:** 4
**Overall Recommendation:** 4
**Confidence:** 4

**Summary:**

This paper proposes TANGO, a graph learning framework motivated by dynamical systems and Lyapunov stability. The idea is to learn a task-driven energy function over node embeddings whose gradient gives an energy-decreasing update direction, and to complement this with a learned tangential flow that moves along energy level sets without changing the energy value. The decomposition is intended to preserve stability while improving flexibility and long-range information propagation.

**Compliance With Llm Reviewing Policy:**

Affirmed.

**Key Questions For Authors:**

1. Can the authors provide more direct empirical evidence that TANGO alleviates oversquashing, beyond better performance on long-range tasks? For example, graph bottleneck diagnostics or feature propagation measures would materially strengthen the main claim.
1. Does the learned energy admit any meaningful post-hoc interpretation across tasks, or is it purely a latent training device?
1. Under what conditions does the tangential component risk dominating the descent component and weakening the intended stability properties in practice?

**Limitations:**

The paper does not adequately discuss limitations. I suggest adding something such as a short section on the indirect nature of the oversquashing evidence, possible sensitivity to architectural choices and discretization depth, the added optimization complexity of learning both an energy and a tangential flow, and the limited interpretability of the learned energy in real applications.

**Strengths And Weaknesses:**

Soundness: The paper is technically thoughtful and empirically broad. The use of a learned energy plus tangential flow is a principled construction. The main weakness is that some central empirical claims, especially around oversquashing mitigation, are still somewhat indirect. Strong results on long-range tasks are encouraging, but I would have liked more direct diagnostics connecting the learned dynamics to reduced oversquashing or better-conditioned information flow.

Presentation: The paper is well written overall, especially in motivating why pure gradient-flow dynamics may be too restrictive. The figures and examples help. Because the method combines several ideas from dynamical systems, energy-based learning, and message passing, some readers may still struggle to map the theory directly to the final architecture.

Significance: The contribution is meaningful because stability and long-range propagation remain central issues in graph learning. A framework that can improve propagation while maintaining a principled stability story is of clear interest, and the breadth of benchmarks increases the paper’s relevance. If the oversquashing interpretation holds more broadly, this could influence future graph dynamical models.

Originality: The proposed combination of a learned Lyapunov-style energy with a tangential, energy-preserving flow is original and conceptually interesting. The work distinguishes itself from prior graph ODE and energy-based GNN formulations by explicitly using both descent and tangential components instead of only one of them.

---

> ### Author Rebuttal · Authors · 2026-03-31
>
> We sincerely thank the reviewer for the positive evaluation of the paper. We appreciate the method being viewed as **technically thoughtful**, **principled**, and **meaningful** for long-range propagation and stability, as well as the **excellent originality score**. We are also grateful for your constructive feedback, which we address below. We hope our responses are satisfactory and that you will consider revising your score.
>
> ---
>
> **1. Re oversquashing alleviation:** Thank you for the suggestion. The original submission includes multiple benchmarks evidencing oversquashing alleviation:
> - Barbell bottleneck diagnostic (Figure 2): a Dirichlet-energy gradient flow struggles to propagate information through the bottleneck, whereas TANGO does so effectively. This is further discussed in Appendix D.2, and we have improved the link between the main text and appendix.
> - Synthetic graph-property prediction tasks: these require long-range propagation, and the benchmark families include structural bottlenecks known to induce oversquashing, as noted in Appendix D.3. Following your comment, we revised the paper to better highlight this discussion.
> - Long-range peptides benchmarks: TANGO is competitive on both Peptides-func and Peptides-struct from LRGB.
>
> **Moreover**, following your suggestion, we added the following diagnostic. We repeated the synthetic barbell experiment on the same topology as Figure 2 under two matched propagation budgets: 50 steps, matching Figure 2, and 100 steps as an additional test on the same graph. For each budget, we compared GCN, a gradient-flow baseline, and TANGO. We measured a structural bottleneck score using pairwise effective resistance, following the topology-based analysis of [1], and a feature-propagation score using a Jacobian-based source-target response, following the sensitivity-based view of oversquashing in [2]. Effective resistance confirms that cross-clique pairs are the hardest communication regime, with mean values 0.20 for same-clique pairs, 1.18 for bridge pairs, and 2.36 for cross-clique pairs. In the table, we report cross-clique transmission (CCT ↑), cross-clique Jacobian response (CJR ↑), and the selected/background Jacobian ratio (SBR ↑), where CCT measures the amount of signal from the source side that reaches the opposite clique after propagation, CJR the mean absolute source-target Jacobian on cross-clique pairs, and SBR the ratio between the Jacobian on task-relevant cross-clique pairs and that on same-clique background pairs. TANGO consistently achieves the strongest propagation on the highest-bottleneck cross-clique pairs under both the 50-step and 100-step budgets across all reported metrics. We added these results to the revised paper. Thank you.
>
> |Budget|Metric|TANGO|Gradient flow|GCN|Gain over gradient flow|Gain over GCN|
> |---|---|---:|---:|---:|---:|---:|
> |50 steps|CCT ↑|0.0508|0.0237|0.0172|+114.3%|+195.3%|
> |50 steps|CJR ↑|0.0745|0.0241|0.0176|+209.1%|+323.3%|
> |50 steps|SBR ↑|1.1229|0.3238|0.2174|+246.8%|+416.6%|
> |100 steps|CCT ↑|0.0564|0.02495|0.01839|+126.1%|+206.7%|
> |100 steps|CJR ↑|0.1370|0.0366|0.0292|+274.3%|+369.2%|
> |100 steps|SBR ↑|1.1487|0.3312|0.2217|+246.8%|+418.1%|
>
> [1] On Over-Squashing in Message Passing Neural Networks: The Impact of Width, Depth, and Topology, ICML 2023.
>
> [2] Understanding Over-Squashing and Bottlenecks on Graphs via Curvature, ICLR 2022.
>
> ---
>
> **2. Re energy interpretation:** Thank you for the insightful question. In TANGO, the learned energy $V_G$ is task-driven, with parameters optimized through the downstream task. Its role is therefore to shape the hidden-state dynamics through the descent and tangential terms. While this work focuses on downstream tasks and introducing TANGO, we believe future work could use the energy function for interpretation, for example in generative models or graph ranking. We added this discussion to the paper. Thank you.
>
> ---
>
> **3. Re tangential term:** In the continuous-time form of TANGO, the tangential term is orthogonal to $\nabla_H V_G$, so by the standard Lyapunov derivative identity, $\frac{d}{dt}V_G(H(t))=\langle \nabla_H V_G(H(t)), \dot H(t)\rangle$ [1], it does not affect the instantaneous energy derivative; dissipation is governed by the descent term via $\alpha_G \ge 0$. Thus, the tangential term preserves flexibility in the feature dynamics while remaining consistent with Lyapunov stability. In practice, we control the balance between the two terms through the learned coefficients $\alpha_G$ and $\beta_G$, and we revised the paper to make this clearer. Thank you for your guidance.
>
> [1] Nonlinear Systems, 2002.
>
> ---
>
> **4. Re limitations:** We appreciate your suggestion and revised the paper to better discuss the limitations, following your guidance. We believe this revision, together with the above discussions that were also added to the paper, helps streamline the path from theory to the final architecture, as noted in your review. Thank you.

---

> > ### Author Rebuttal · Reviewer_YJEt · 2026-04-04
> >
> > The added oversquashing diagnostics provide more direct evidence beyond performance and significantly strengthen the central claim. The explanation of the tangential term and its role in preserving Lyapunov stability while improving flexibility is clear and technically sound. The discussion on energy interpretation is reasonable but still limited, and the learned energy remains largely a latent construct at this stage. I also appreciate the added limitations section, which improves balance and clarity.
> >
> > Overall, it increase my confidence in both the empirical support and positioning, reinforcing my weak accept.

---

> > > ### Author Response · Authors · 2026-04-04
> > >
> > > Dear Reviewer YJEt,
> > >
> > > We thank you for your detailed and actionable review, and for your rebuttal acknowledgement. We believe that your feedback is beneficial to the quality of our work.
> > >
> > > We are happy to read that you found our added diagnostics to significantly strengthen the central claim of our work, the explanation of the tangential term to be sound, and your appreciation of the proposed changes to the paper, which we confirm have been incorporated into our paper.
> > >
> > > Lastly, we would like to thank you for maintaining and reinforcing your positive view of our work and your recommendation of the acceptance of our work.
> > >
> > >
> > > With warmest regards, \
> > > Authors.

---

### Official Review · Reviewer_31cB · 2026-03-18

**Soundness:** 2
**Presentation:** 2
**Significance:** 2
**Originality:** 2
**Overall Recommendation:** 4
**Confidence:** 2

**Summary:**

The paper introduces a new graph representation learning framework that handles the node feature dynamics by a combination of energy gradients flows, and tangential flows. The flexible representation of the graph dynamics enables more efficient learning, especially in ill-conditioned regions of the loss landscape. The proposed representation is combined with different GNNs backbones, and yields competitive performance on graph benchmarks.

**Compliance With Llm Reviewing Policy:**

Affirmed.

**Final Justification:**

relatively convincing response. Still not fully clear on the actual relevance of the general theoretical results, but this is not extremely critical. Score can be raised a bit.

**Key Questions For Authors:**

see weaknesses above.

**Limitations:**

yes

**Strengths And Weaknesses:**

Strengths:
1. The paper presents TANGO, a novel representation framework in graph learning, building on ideas for improving energy functions that have been proposed in deep learning, or different graph learning settings. Its specific version adapted to the learning of the energy function, that is driven by specific graph tasks, is novel.
2. The paper connects the novel representation framework to theoretical results in dynamical systems and stability, which provides a solid basis for the proposed contribution.
3. Numerous experiments, and rather detailed discussions, offer a relatively convincing perspectives of the benefits offered by TANGO. Generally, the performance are shown to be on-par, or slightly better than state-of-the-art methods in node or graph classification and regression benchmarks,

Weaknesses:
1. The presentation is generally clear, but the text could be improved by removing redundancies (see end of introduction, or multiple repetitions of related works)
2. Important information related to the implementation of the learning framework, notably the relative weights of the different components of the energy function, are not discussed in details,
3. Results are generally convincing, but a more structured presentation of the experimental analysis would be helpful. For example, many factors are inter-twinned, like tasks, datasets, GNN backbones, competing methods - and it is not easy to really appreciate in which settings the proposed framework brings most of its benefits, and the reasons for that.
4. The effort in developing theoretical results is good, but generally do not bring much core results related to the actual framework and use cases. For example, 'This implies that in contrast to gradient flows, the dynamics of TANGO obtained by the tangential term can evolve even in regions where the energy landscape is flat.' on page 5 is pretty much expected. Instead, the important benefits on alleviating over-squashing, for example, is only studied from an empirical perspective? Or the benefits of combinations with specific GNN backbones is not discussed from a theoretical perspective. Those would provide stronger arguments for the proposed framework, than the few interesting, but expected, theoretical results in the paper.

While the paper looks very exciting at first, it may fall a bit short of new actual results, and lack responses to key questions. The work stays promising however.

---

> ### Author Rebuttal · Authors · 2026-03-31
>
> We sincerely thank you for highlighting several central strengths of the paper, including that TANGO is a **novel representation framework in graph learning**, that the dynamical-systems analysis provides a **solid basis** for the contribution, and that the experiments offer a **convincing perspective** on the benefits of the method. We also appreciate your observation that the work is **exciting** and **promising**. We welcome your constructive feedback, which we address in detail below, and hope that you find our responses satisfactory and will consider revising your score.
>
> ---
>
> **1. Re presentation:** Thank you for your helpful suggestion. We have revised the paper and streamlined the introduction and related work sections, and we think it helps to improve the presentation of our work.
>
> ---
>
> **2. Re implementation details and relative weights of the components:**
> We note that as described in Section 3.1 we draw the blueprint of TANGO, and in Section 3.2 we discuss specific implementation details of the components of TANGO, including the balance between the terms: the TANGO update is $H^{(\ell+1)} = H^{(\ell)} + \epsilon \left(-\alpha_G(H^{(\ell)}) \nabla_H V_G(H^{(\ell)}) + \beta_G(H^{(\ell)}) T_{V_G}(H^{(\ell)}) \right)$, where $\alpha_G$ and $\beta_G$ are learned as described in Equation (8) and Equation (10), respectively. We also define the learned energy in Equation (7), and the learned tangential term in Equation (9).  We have also included detailed descriptions of our experimental settings related to the implementation in Appendix D. Following your comment, we have revised the paper to expand the discussion and better connect the main text with the Appendix. Thank you.
>
> ---
>
> **3. Re structure of the experiments section:** We thank you for your thoughtful comment. This is an aspect that we also considered in our paper. The submission organizes the experiments by benchmark family and purpose, covering synthetic long-range graph-property tasks, real-world long-range interaction on Peptides, standard graph benchmarks, and heterophilic node classification, and evaluates TANGO across four backbones (GCN, GIN, GatedGCN, GPS), with Appendix D detailing the benchmark setup and baselines and Appendix E providing depth and component ablations. Following your suggestion, we revised the presentation to make this empirical story clearer: we now state more directly where TANGO helps most and why. In particular, we clarify that the largest gains appear on the synthetic graph-property tasks, which directly stress long-range propagation and bottlenecks; that TANGO remains highly competitive on LRGB and achieves the best result on Peptides-struct; that on standard graph benchmarks it consistently improves its base backbones while remaining competitive with strong general-purpose methods; and that it also remains strong in heterophilic settings while consistently improving its backbone models there as well. We also make the backbone-level story more explicit across GCN, GIN, GatedGCN, and GPS, so that it is easier to see both where the gains are largest and how they align with the motivation behind TANGO. Thank you for your guidance, which helped us improve the presentation of our results.
>
> ---
>
> **4. Re theory and practice:** We thank you for the important comment. In the revision we make the role of the theory more explicit in relation to the framework. Specifically, we clarify that that: Proposition 4.1 establishes Lyapunov dissipation of the learned energy, Proposition 4.2 shows that the tangential component enables continued feature evolution in flat regions, Proposition 4.3 identifies ill-conditioning as the regime where pure gradient descent becomes ineffective, and Proposition 4.4 shows that the combined TANGO direction can realize improved update directions in such regimes. We then connect this directly to graph learning by making the Section 4 link between bottlenecks, poor conditioning, and oversquashing more explicit, and by referring more clearly to Figure 2, where the badly conditioned barbell graph provides a concrete downstream example of this mechanism in action. We also clarify the backbone-level story: because TANGO modifies the learned dynamics at the framework level, the theory explains why Lyapunov-guided descent and tangential flow should be broadly beneficial, and the experiments confirm this consistently across GCN, GIN, GatedGCN, and GPS, with the strongest gains in long-range and bottleneck-sensitive settings. To make this easier to assess, we revised the presentation so that this theory-to-use-case bridge appears directly in the main text and is reinforced with a targeted bottleneck / feature-propagation diagnostic. We also added diagnostics in our response to point 1 to Reviewer  YJEt and point 1 to Reviewer BFpR, that further connect our theory and practice.  Thank you for your thoughtful suggestions.

---

> > ### Author Rebuttal · Reviewer_31cB · 2026-04-02
> >
> > Thanks for the thoughtful and relatively convincing rebuttal. The relevance and validity of the theoretical results is still not fully convincing, with respect to the actual settings in practice (many components cannot be modelled faithfully in the proposed theoretical framework). In any case, the trends supported by theory, seem to be generally confirmed in the experiments, which is certainly positive (but not fully conclusive wrt the relevance of the general theoretical results proposed in the paper. In any case, if the proposed changed are integrated in the revised version, the score can be increased.

---

> > > ### Author Response · Authors · 2026-04-04
> > >
> > > Dear Reviewer 31cB,
> > >
> > > We thank you for your constructive and actionable review, and for your rebuttal acknowledgement. Your feedback is beneficial to the quality of our work, in our opinion, and we would like to confirm that we have included the discussions, clarifications, and added results, to the revised paper.
> > >
> > > We are glad to read that you found our responses thoughtful and convincing. We would like to thank you for increasing your score and recommending the acceptance of our work.
> > >
> > >
> > > With kindest regards, \
> > > Authors.

---

### Decision · Program_Chairs · 2026-04-30

**Decision:**

Accept (regular)

**Comment:**

Reviewers agreed that the learned energy plus tangential-flow decomposition is original, well motivated, and supported by strong experimental coverage. The main concerns were that some theoretical claims were more general than directly actionable, and that the connection between the learned dynamics and the claimed oversquashing benefits could be made more explicit. The rebuttal added useful diagnostics and clarifications, which substantially strengthened reviewer confidence without revealing a core correctness problem. Overall, I recommend Accept because the paper combines conceptual novelty, technical quality, and broad empirical evidence in a way that should make it a solid contribution to ICML.